# Exportin-mediated nucleocytoplasmic transport maintains Pch2 homeostasis during meiosis

**Esther Herruzo**[1], **Estefanía Sánchez-Díaz**[1], **Sara González-Arranz**[1], **Beatriz Santos**[1,2], **Jesús A. Carballo**[1], **Pedro A. San-Segundo**[1]*

**1** Instituto de Biología Funcional y Genómica (IBFG), CSIC-USAL, Salamanca, Spain, **2** Departamento de Microbiología y Genética. University of Salamanca. Salamanca, Spain

* pedross@usal.es

**Data Availability Statement:** The relevant data are contained within the manuscript. Numerical data underlying the graphs are available in S1 and S2

## Abstract

The meiotic recombination checkpoint reinforces the order of events during meiotic prophase I, ensuring the accurate distribution of chromosomes to the gametes. The AAA+ ATPase Pch2 remodels the Hop1 axial protein enabling adequate levels of Hop1-T318 phosphorylation to support the ensuing checkpoint response. While these events are localized at chromosome axes, the checkpoint activating function of Pch2 relies on its cytoplasmic population. In contrast, forced nuclear accumulation of Pch2 leads to checkpoint inactivation. Here, we reveal the mechanism by which Pch2 travels from the cell nucleus to the cytoplasm to maintain Pch2 cellular homeostasis. Leptomycin B treatment provokes the nuclear accumulation of Pch2, indicating that its nucleocytoplasmic transport is mediated by the Crm1 exportin recognizing proteins containing Nuclear Export Signals (NESs). Consistently, leptomycin B leads to checkpoint inactivation and impaired Hop1 axial localization. Pch2 nucleocytoplasmic traffic is independent of its association with Zip1 and Orc1. We also identify a functional NES in the non-catalytic N-terminal domain of Pch2 that is required for its nucleocytoplasmic trafficking and proper checkpoint activity. In sum, we unveil another layer of control of Pch2 function during meiosis involving nuclear export via the exportin pathway that is crucial to maintain the critical balance of Pch2 distribution among different cellular compartments.

## Author summary

Meiosis is a specialized cell division essential for sexual reproduction because it is responsible for halving the number of chromosomes during gametogenesis. Meiosis involves a series of events that culminate in the interaction between maternal and paternal chromosomes, and the exchange of genetic material by recombination. These processes not only generate genetic diversity, but also, and more important, are essential to promote proper distribution of chromosomes to the gametes. Meiotic cells possess surveillance mechanisms, called checkpoints, that reinforce the proper order of events during the meiotic cell cycle, ensuring the accurate segregation of chromosomes and preventing the formation of

Files. The sequences of all primers used in strain and plasmid construction are available in S3 File.

**Funding:** This work was supported by grant PID2021-125830NB-I00 from Ministry of Science and Innovation (MCIN/AEI/10.13039/501100011033/) and "FEDER Una manera de hacer Europa" to PASS and JAC. EH was partially supported by the grant CSI259P20 from the "Junta de Castilla y León" (co-funded by FEDER). ESD is supported by a predoctoral contract from the "Junta de Castilla y León" (co-funded by the Education Department and the European Social Fund FSE+). The funders had no role in study design, data collection and analysis, decision to publish, or preparation of the manuscript.

aneuploid gametes. The Pch2 protein is an evolutionarily-conserved ATPase that functions in the yeast meiotic recombination checkpoint. Pch2 localizes in the nucleolus, chromosomes, and the cytoplasm, performing distinct and, in some cases, opposite functions in these different subcellular compartments. In this work, we uncover the mechanism that Pch2 uses to travel from the nucleus to the cytoplasm; in particular, the Crm1 exportin pathway. We also identify and characterize a nuclear export signal in the non-catalytic domain of Pch2 required for its nucleocytoplasmic trafficking. Thus, our work contributes to understanding how the critical balance of Pch2 subcellular distribution is achieved to support faithful meiotic completion.

## Introduction

In sexually-reproducing organisms, chromosome distribution to the gametes relies on a specialized type of cell division, called meiosis, which reduces the number of chromosomes to the half prior to fertilization [1]. During the long meiotic prophase I, chromosomes undergo several elaborate and carefully-regulated processes: pairing, recombination and synapsis. Recombination involves the formation of developmentally programmed DNA double-strand breaks (DSBs) by the conserved Spo11 protein and its partners [2]. The interhomolog repair of a subset of these DSBs as crossovers (COs) is essential for accurate chromosome segregation [3–5]. This process requires chromosomes to undergo intimate interactions during meiotic prophase that are facilitated by active chromosome movements coordinated by cytoskeletal forces and the widely conserved linker of nucleoskeleton and cytoskeleton (LINC) complex [6–8]. Once each chromosome has paired with its homologous partner, a tripartite structure called the synaptonemal complex (SC) is assembled along the length of each chromosome pair to maintain a stable association between paired homologs. In *Saccharomyces cerevisiae*, Hop1, Red1 and Rec8 are structural components of the axial cores of the SC [9–12], and the transverse filament Zip1 protein composes the central region which holds together the axes [13]. The central region also includes the central element made of the Ecm11 and Gmc2 proteins [14]. The axial elements are referred to as lateral elements when the central region is assembled, and the SC is fully formed. In the *zip1Δ* mutant, lacking the central region, paired homologous partners are still connected by axial associations formed at crossover sites [12]. The fidelity of meiotic divisions is ensured by the meiotic recombination checkpoint, which blocks cell cycle progression at the end of prophase I while recombination intermediates (i.e., unrepaired DSBs) persist [15]. Briefly, the Mec1-Ddc2 sensor kinase is recruited to resected DSBs [16] and phosphorylates Hop1 at various consensus S/T-Q sites in a Red1-dependent manner. Among the multiple S/T-Q sites in Hop1, phosphorylation of T318 (hereafter, Hop1-T318ph) is critical for checkpoint activation [17–19]. The Pch2 AAA+ ATPase sustains adequate levels of Hop1-T318ph on chromosomes to signal checkpoint activity in *zip1Δ* [20]. Hop1-T318ph, in turn, promotes the activation of Mek1 and its recruitment to chromosomes [21–23]. Finally, the Mek1 effector kinase phosphorylates and inhibits the Ndt80 transcription factor leading to pachytene arrest [24]. In addition, Mek1 acts on Rad54 and Hed1 helping to prevent intersister recombination [25, 26].

Pch2 (known as TRIP13 in mammals) is a member of the AAA+ ATPase family; these AAA+ proteins use the energy provided by ATP hydrolysis to provoke conformational changes on their substrates [27, 28]. The conserved Pch2 protein was initially identified in a screen for meiotic recombination checkpoint mutants in budding yeast [29], but in addition to the checkpoint it also participates in multiple meiotic processes regulating several aspects of

CO recombination and chromosome morphogenesis both in yeast and other organisms [30–37]. Despite the varying effects resulting from the absence of Pch2 in different organisms, it has been proposed that the common meiotic function of Pch2 is the coordination of recombination with chromosome synapsis to ensure proper CO number and distribution [38].

The meiotic roles of Pch2 are exerted through its action on its preferred client: the HORMAD protein Hop1 [39, 40]. As a member of the HORMAD family, Hop1 contains a flexible safety belt in its HORMA domain, which enables it to adopt an open/unbuckled or closed conformation [41, 42]. The transition from closed-Hop1 to unbuckled-Hop1 is thought to be accomplished by Pch2 ATPase activity, poising Hop1 for binding to a closure motif in Hop1 itself, or in other proteins that interact with Hop1, like Red1[42–47]. In other organisms, Pch2$^{TRIP13}$ requires the p31(COMET) adaptor, but no cofactor has yet been described for Pch2 action in yeast [48–51].

Pch2 localization studies feature a complicated scenario. Pch2 is highly enriched in the unsynapsed ribosomal DNA (rDNA) region. This rDNA-specific recruitment requires Orc1, which collaborates with Pch2 to exclude Hop1 from the nucleolus thus limiting meiotic DSB formation at the repetitive rDNA array [29, 52]. Pch2 is also detected as individual foci colocalizing with Zip1 on synapsed chromosomes [29, 32]. Targeting of Pch2 to the SC depends on Zip1, but other factors such as RNAPII-dependent transcription [53], Top2 [54], Nup2 [55] or chromatin modifications driven by Sir2 and Dot1 [23, 56, 57], also influence Pch2 chromosomal distribution. Pch2 recruitment to chromosomes removes Hop1 from the axes [20, 58, 59], likely by disrupting Hop1-Red1 interaction via its remodeling activity towards the HORMA domain [41]. Hop1 removal downregulates DSB formation on chromosomes that have successfully identified their homologs and formed COs, and silences the meiotic recombination checkpoint [60, 61]. In addition to the nucleolar and chromosomal population, a cytoplasmic supply of Pch2 also exists; this cytoplasmic pool is necessary and sufficient to sustain meiotic checkpoint activation [60, 62]. Based on all these observations, and the demonstrated role of Pch2 in promoting Hop1 chromosomal incorporation and T318 phosphorylation in *zip1Δ* [20], a model emerges in which Pch2, from the cytoplasm, structurally remodels Hop1 by using its ATPase activity, ensuring that unbuckled Hop1 is available to associate with Red1 via the closure motif determining its axial incorporation and achieving proper phosphorylation levels at T318 to support checkpoint activity [60].

The importance of a precise localization of Pch2 for an accurate meiotic checkpoint response argues that its distribution between the different cellular compartments must be finely regulated. Here, we combine detailed cytological, molecular and genetics studies to provide novel insights into how Pch2 travels from the nucleus to the cytoplasm, thereby contributing to delineate the regulatory network governing Pch2 subcellular distribution and function.

## Results and discussion

### Pch2 nuclear export is mediated by the Crm1 exportin

The existence of at least three different subpopulations of Pch2 residing in the nucleolus, chromosomes and cytoplasm, together with the recent observation that an exquisite balance of Pch2 subcellular distribution is crucial to maintain a proper meiotic recombination checkpoint response [60], suggest that the nucleocytoplasmic transport of Pch2 must be tightly controlled. As an initial approach to explore this mechanism, we analyzed if Pch2 travels from the nucleus to the cytoplasm via Crm1, which is the main exportin mediating the nuclear export of proteins containing a Nuclear Export Signal (NES) [63]. The drug leptomycin B (LMB) is a powerful tool to establish whether the nuclear export of a NES-containing protein is mediated by the CRM1/XPO1/KAP124 exportin pathway. LMB binds to CRM1 disrupting the

formation of the trimeric NES-CRM1-RanGTP export complex required for the transport from the nucleus to the cytoplasm [64]. Thus, LMB treatment leads to the nuclear accumulation of NES-containing proteins. In *Schizosaccharomyces pombe* and mammalian cells, LMB is capable of binding directly to CRM1 inhibiting nuclear export, but *S. cerevisiae* cells are resistant to LMB because the wild-type budding yeast Crm1 does not bind to the drug. However, the Crm1-T539C mutant version is capable of binding LMB with high affinity and renders *S. cerevisiae* cells sensitive to LMB [65]. Thus, to determine whether Pch2 travels from the nucleus to the cytoplasm via the Crm1 pathway, we analyzed Pch2 subcellular localization after LMB treatment in a *crm1-T539C* mutant background (Fig 1A–1D). First, we confirmed that control *crm1-T539C* diploid cells, in the absence of LMB, showed normal sporulation efficiency and completed meiotic divisions, albeit at slightly reduced efficiency and slower kinetics compared with the *CRM1* control (S1A and S1B Fig). However, addition of LMB during prophase I (15h) blocked meiotic progression (S1B Fig), indicating that, as expected, nuclear export is required for completion of meiosis and sporulation. More important, we also checked that the untreated *zip1Δ crm1-T539C* mutant showed a strong meiotic arrest (S1A and S1B Fig), indicating that the *crm1-T539C* mutation itself, in the absence of LMB, does not alter the *zip1Δ*-induced checkpoint response. Thus, the *crm1-T539C* mutant is a valid tool to explore the implication of nuclear export in Pch2 subcellular distribution in *S. cerevisiae*.

Pch2 localization was assessed by fluorescence microscopy analysis of a functional version of GFP-tagged Pch2 expressed from the previously described $P_{HOP1}$-GFP-PCH2 construct [60, 62], referred to as *GFP-PCH2* throughout the article for simplicity. To avoid side effects resulting from differences in meiotic progression, GFP-Pch2 subcellular distribution was studied in prophase I-arrested *ndt80Δ crm1-T539C* live meiotic cells in otherwise wild-type (*ZIP1)* and *zip1Δ* backgrounds. Cells were treated with LMB, or mock-treated with the solvent ethanol, as control. In mock-treated wild-type *ZIP1* cells, Pch2 localized to one side of the nucleus in a region that was previously shown to correspond to the nucleolus, it was also detected as discrete faint nuclear foci corresponding to the synapsed chromosomes, and it showed diffuse homogenous cytoplasmic signal as well (Fig 1B, top panels) [60, 62]. Addition of LMB led to a strong accumulation of Pch2 in the nucleus displaying a much more intense signal especially in the presumed nucleolus, but also on the chromosomes and even in the nucleoplasm (Fig 1C, top panels). On the other hand, in the *zip1Δ* untreated control, Pch2 was only present in the nucleolus and the cytoplasm, as previously described (Fig 1B, bottom panels) [60, 62], but LMB addition resulted in a prominent nucleolar accumulation and nucleoplasmic distribution, concomitant with reduced cytoplasmic localization (Fig 1C, bottom panels). Accordingly, the ratio between nuclear (including nucleolus) and cytoplasmic GFP signal increased after LMB treatment both in *ZIP1* or *zip1Δ* strains (Fig 1D). Despite the altered subcellular distribution, total GFP-Pch2 protein levels remained unchanged in the presence or absence of LMB (Fig 1E).

To obtain more detailed information, we also analyzed the localization of GFP-Pch2 by immunofluorescence on pachytene chromosome spreads after LMB treatment. We first describe the distribution pattern in *ZIP1* strains. In the absence of LMB, and consistent with the well-characterized localization of Pch2 [20, 29, 32] and the observations in live meiotic cells (Fig 1B), the GFP-Pch2 protein localized mainly in the rDNA region, marked by the nucleolar protein Nsr1 (Fig 1F). The chromosome-associated population of Pch2, which is only present in synapsis-proficient strains, was not easily detectable with this technique in the BR strain background, as previously reported [20, 62] (Fig 1F). However, the addition of LMB led to a general increase of GFP-Pch2 amount on the spread nuclei (Fig 1H), affecting both the nucleolar (Fig 1I) and the chromosomal (Fig 1J) populations. Moreover, in LMB-treated *ZIP1* cells, the accumulation of Pch2 in the nucleus was also associated with the increased formation

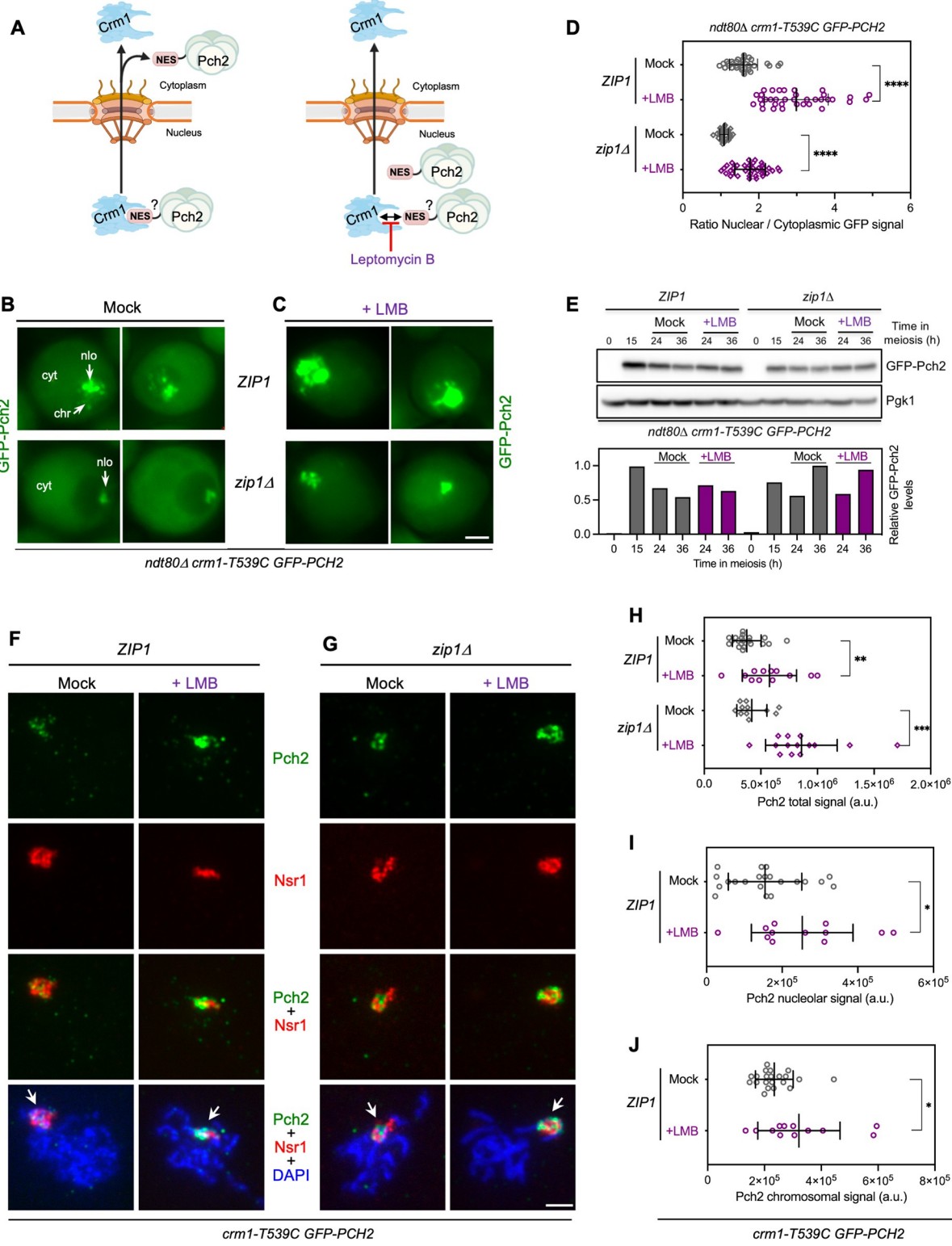

**Fig 1. Nuclear export of Pch2 is blocked by leptomycin B (LMB). (A)** Schematic representation of protein nuclear export via nuclear pore complexes mediated by the interaction of a Nuclear Export Signal (NES) with the Crm1 exportin (see text for details). Leptomycin B (LMB) blocks NES binding to Crm1 preventing nuclear export. The presence of a putative NES in Pch2 is depicted. For simplicity, the NES is drawn only in one subunit of the Pch2 hexamer. BioRender.com was used to create this figure. **(B, C)** Fluorescence microscopy images of GFP-Pch2 localization in *ZIP1* and *zip1Δ* cells, mock-treated (B), or treated with 500 ng/ml LMB (C), 15 h after meiotic induction. Images were taken at

24 h. Two representative individual cells for each condition are shown. Localization of GFP-Pch2 in the nucleolus (nlo), chromosomes (chr), and cytoplasm (cyt) is indicated. Scale bar, 2 μm. Strains are: DP1927 (*ndt80Δ crm1-T539C GFP-PCH2*) and DP1837 (*zip1Δ ndt80Δ crm1-T539C GFP-PCH2*). **(D)** Quantification of the ratio of nuclear (including nucleolar) to cytoplasmic GFP fluorescent signal for the experiment shown in (B, C). Error bars, SD. **(E)** Western blot analysis of GFP-Pch2 production (using anti-Pch2 antibodies) in the absence (mock) or presence of LMB. Pgk1 was used as loading control. The graph shows the quantification of GFP-Pch2 relative levels. **(F, G)** Immunofluorescence of spread meiotic chromosomes at pachytene stained with anti-Pch2 antibodies (to detect GFP-Pch2; green), anti-Nsr1 antibodies (red) and DAPI (blue). Representative *ZIP1* (F) and *zip1Δ* (G) nuclei, mock-treated, or treated with 500 ng/ml LMB 15 h after meiotic induction are shown. Arrows point to the rDNA region. Spreads were prepared at 19 h. Scale bar, 2 μm. Strains are: DP1717 (*crm1-T539C GFP-PCH2*) and DP1721 (*zip1Δ crm1-T539C GFP-PCH2*). **(H)** Quantification of the GFP-Pch2 total signal for the experiment shown in (F, G). Error bars, SD; a.u., arbitrary units. **(I-J)** Quantification of the nucleolar (I) and chromosomal (J) GFP-Pch2 signal for the experiment shown in (F). Error bars, SD; a.u., arbitrary units.

of polycomplexes, which are organized extrachromosomal assemblies of SC components [66], in 29.4% of nuclei (S2A and S2C Fig). On the other hand, on spread chromosomes of the *zip1Δ* mutant, Pch2 is only associated with the rDNA region; the addition of LMB led to an increased GFP-Pch2 signal restricted to the nucleolar area (Fig 1G and 1H).

In sum, these results demonstrate that inhibition of the Crm1 exportin pathway leads to Pch2 nuclear accumulation indicating that Pch2 travels from the nucleus to the cytoplasm using this pathway.

## Effect of Crm1-mediated export inhibition on checkpoint function

We have previously described that the main role of Pch2 in the *zip1Δ*-induced checkpoint is to promote Hop1 association to unsynapsed chromosome axes supporting high levels of Hop1 phosphorylation at T318, which in turn sustain Mek1 activation [20]. Artificial redirection of Pch2 to different subcellular compartments (i.e., forced nuclear accumulation) impairs these functions [60]. To determine whether inhibition of the Crm1 nuclear export pathway, which alters Pch2 subcellular distribution, affects checkpoint function in *zip1Δ*, we examined the impact on Hop1 incorporation onto chromosome axes as well as the checkpoint status using Hop1-T318 phosphorylation as marker for Pch2-dependent checkpoint activity [17, 19, 20].

Analysis of Hop1 localization by immunofluorescence of *zip1Δ* spread nuclei in the absence of LMB revealed its characteristic intense and continuous signal along the unsynapsed axes, and its exclusion from the rDNA region containing Pch2 (Fig 2A). In contrast, after LMB treatment, Hop1 localization was compromised displaying reduced intensity (Fig 2A and 2B). To circumvent the possible effect of the different kinetics of meiotic progression, emphasized by the meiotic block conferred by the addition of LMB (S1 Fig), we used prophase I-arrested *ndt80Δ* strains for an accurate analysis of Hop1-T318 phosphorylation levels without interference from cell cycle progression. As observed in Fig 2C, in the absence of LMB, the checkpoint was active in *zip1Δ* cells, as manifested by the high levels of Hop1-T318ph, compared to the wild type (*ZIP1*). However, phosphorylation levels of this checkpoint marker, which reflects the DNA damage response to meiotic DSBs, were notably reduced by the addition of LMB in *zip1Δ* cells (Fig 2C).

Thus, the accumulation of Pch2 in the nucleus of *zip1Δ* cells imposed by inhibition of Crm1 correlates with defects in Hop1 localization and phosphorylation, and inactivation of the meiotic recombination checkpoint. We cannot rule out that the nuclear accumulation of other factors involved in the checkpoint besides Pch2 could be responsible for the defective response observed after LMB treatment. For example, nucleocytoplasmic transport also appears to be involved in Ndt80 regulation [67]. Nonetheless, we point out that the forced nuclear accumulation of Pch2 alone by fusion to a strong nuclear localization signal (NLS) leads to checkpoint downregulation [60], arguing that simply blocking Pch2 nuclear export by itself may account for the impaired checkpoint activity observed in LMB-treated *zip1Δ* cells. It is also conceivable

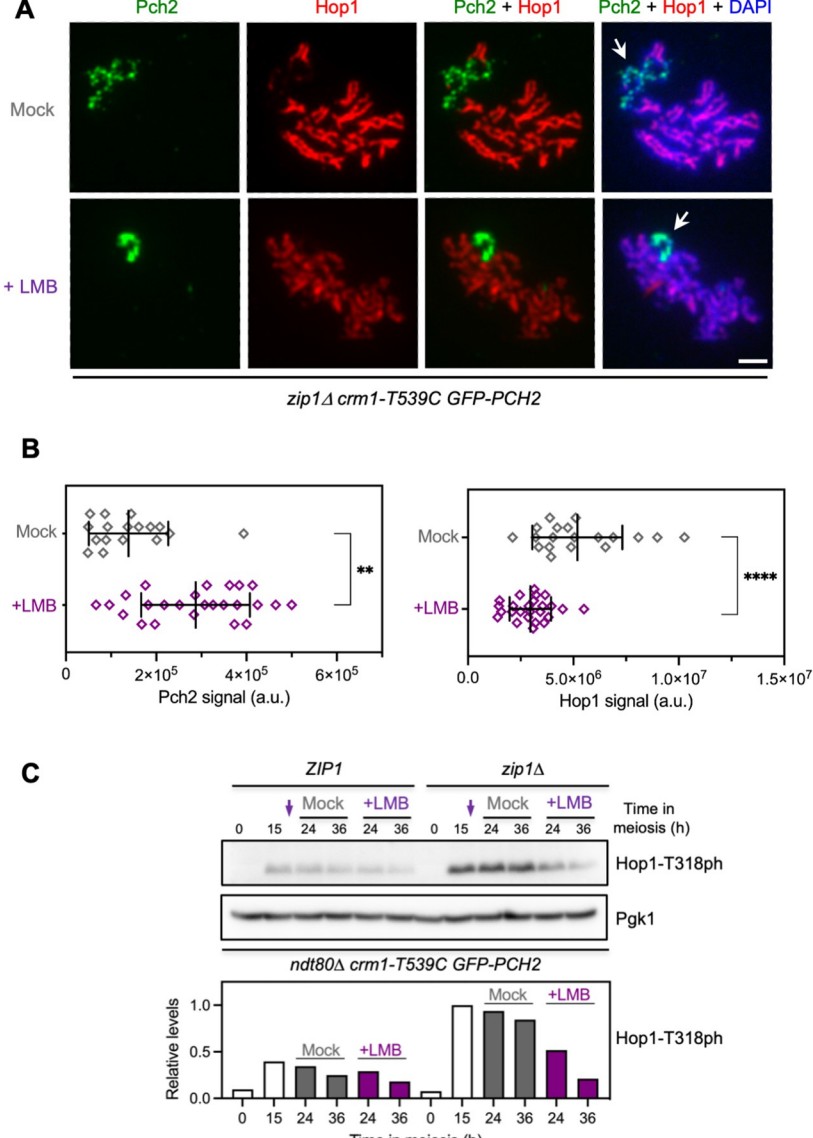

**Fig 2. LMB treatment impairs the meiotic recombination checkpoint. (A)** Immunofluorescence of *zip1Δ* spread meiotic chromosomes stained with anti-Hop1 (red) and anti-GFP (to detect GFP-Pch2; green) antibodies. Representative nuclei, either mock-treated, or treated with 500 ng/ml LMB 15 h after meiotic induction, are shown. Arrows point to the rDNA region. Spreads were prepared at 19 h. Scale bar, 2 μm. The strain is DP1721 (*zip1Δ crm1-T539C GFP-PCH2*). **(B)** Quantification of Pch2 (left graph) and Hop1 (right graph) signal for the experiment shown in (A). Error bars, SD; a.u., arbitrary units. **(C)** Western blot analysis of Hop1-T318 phosphorylation (ph) in the absence (mock) or presence of LMB added at 15 h after meiotic induction (arrows). Pgk1 was used as loading control. The graph shows the quantification of Hop1-T318ph relative levels. Strains are: DP1927 (*ndt80Δ crm1-T539C GFP-PCH2*) and DP1837 (*zip1Δ ndt80Δ crm1-T539C GFP-PCH2*).

that the reduced amount of chromosomal Hop1 when Pch2 accumulates in the nucleus might result in reduced DSB levels that would produce diminished checkpoint signaling.

## Pch2 nucleocytoplasmic traffic is independent of Zip1 and Orc1

Since Orc1 is required for Pch2 nucleolar targeting [52], to further delineate the requirements for Pch2 nucleocytoplasmic transport we analyzed the effect of depleting Orc1 under LMB

treatment conditions. That is, we explored if Pch2 nucleocytoplasmic traffic involves its transit through the rDNA region. To this end, we used the *orc1-3mAID* degron allele previously described; in this mutant, Pch2 does not localize to the nucleolus because of Orc1 degradation induced by auxin [62]. We studied the localization of GFP-Pch2 in *orc1-3mAID* meiotic live cells and on chromosome spreads of both *ZIP1* and *zip1Δ* strains. All the experiments involving *orc1-3mAID* were performed with auxin added at 12 h after meiotic induction to induce Orc1 depletion during prophase I. For clarity, we first describe the localization patterns in *ZIP1* cells. In most mock-treated cells of the *orc1-3mAID* mutant, GFP-Pch2 was only detected in the cytoplasm and on nuclear foci or lines of different sizes likely corresponding to the chromosomes (Figs 3A, mock panel and S2D). Indeed, immunofluorescence analyses of nuclear spreads confirmed the lack of Pch2 in the nucleolus (marked by Nsr1) and its presence on the chromosomes displaying a dotty-linear signal in the LMB-untreated control (Fig 3C, mock panels). In contrast, in the majority of LMB-treated *ZIP1* cells, the cytoplasmic signal was reduced and GFP-Pch2 displayed a robust nuclear localization with accumulation in a strong large focus that could be combined with a dotty or linear pattern (Figs 3A, 3B and S2D). In a small fraction of cells (16–20%), a diffuse nuclear signal could also be found in the presence or absence of LMB. This minor pattern appeared to be exclusive of *orc1-3mAID crm1-T539C* cells (S2D Fig). Analyses of nuclear spreads revealed that, in the presence of LMB, the GFP-Pch2 signal was much more intense, showing a more continuous linear chromosomal pattern (Fig 3C, left LMB panel; Fig 3E). Moreover, 45% of the spread nuclei showed a very intense Pch2 focus that did not correspond to the nucleolus (Fig 3C, right LMB panels), but colocalized with Zip1 at polycomplexes (S2B and S2C Fig). In some cases, the chromosomal staining of Zip1 and Pch2 was somewhat masked by the strong signal of the polycomplex. Thus, we conclude from these results that the block of nucleocytoplasmic traffic in Orc1-depleted cells, lacking nucleolar Pch2, leads to a stronger association of Pch2 with the SC or assemblies of SC components. Next, to determine whether the nuclear retention of Pch2 upon LMB treatment requires its interaction with Zip1, we analyzed Pch2 localization in the *zip1Δ orc1-3mAID* mutant. We observed that GFP-Pch2 localized exclusively in the cytoplasm in the absence of LMB (Fig 3A, mock panel) but, upon LMB treatment, Pch2 was retained in the nucleus displaying a diffuse nucleoplasmic distribution (Fig 3A, LMB panel, Fig 3B). There was no chromatin-associated Pch2 signal in either untreated or LMB-treated nuclei (Fig 3D and 3E). In sum, these observations suggest that shuttling of Pch2 between the nucleus and the cytoplasm involves neither the SC (Zip1-dependent) nor the nucleolar (Orc1-dependent) association of Pch2. We conclude that, although Orc1 and Zip1 clearly affect the intracellular localization of Pch2 since its nuclear/cytoplasmic ratio is altered in the absence of Orc1 and/or Zip1 [60] (Fig 1D and 3B), the Crm1-mediated transport of Pch2 per se does not entail Orc1 or Zip1.

## A nuclear export signal in the amino-terminus of Pch2 promotes export out of the nucleus

The Crm1 exportin binds proteins possessing Nuclear Export Signals (NESs) for its transport out of the nucleus [68]. We have observed that Pch2 export depends on Crm1 but, in principle, Pch2 could travel bound to putative partner(s) containing NESs or via NES(s) present in Pch2 itself. Thus, we searched for consensus NESs in the Pch2 sequence using the LocNES prediction tool [69] combined with visual matching to known patterns for hydrophobic residues distribution in NESs [70]. Three putative NESs with high score probability were predicted; all of them present in the N-terminal domain (NTD) of Pch2 at amino acid positions 98–107, 127–136 and 205–214. To determine whether these putative NESs are functionally relevant for Pch2 nucleocytoplasmic traffic, we generated three GFP-tagged Pch2 versions harboring

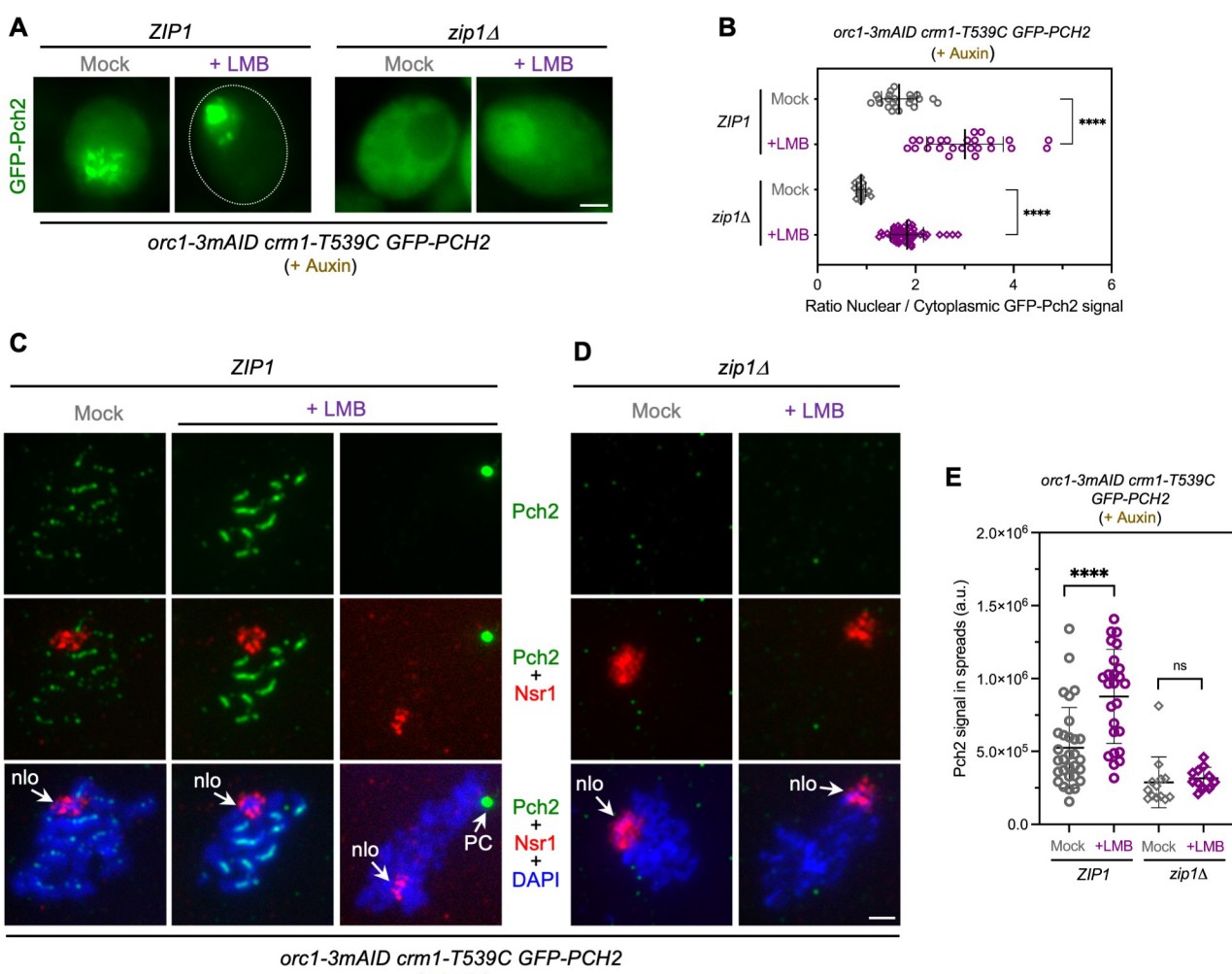

**Fig 3. Pch2 nucleocytoplasmic traffic does not involve its association with Orc1 and Zip1. (A)** Fluorescence microscopy images of GFP-Pch2 localization in *ZIP1 orc1-3mAID* and *zip1Δ orc1-3mAID* cells, mock-treated, or treated with 500 ng/ml LMB, 15 h after meiotic induction. Auxin (500μM) was added to the cultures 12 h after meiotic induction to induce Orc1 degradation. Images were taken at 19 h. Cells representing the predominant GFP-Pch2 localization pattern for each condition are shown. Additional examples of GFP-Pch2 distribution are presented in S2D Fig. Scale bar, 2 μm. Strains are: DP1885 (*orc1-3mAID crm1-T539C GFP-PCH2*) and DP1886 (*zip1Δ orc1-3mAID crm1-T539C GFP-PCH2*). **(B)** Quantification of the ratio of nuclear (including nucleolar) to cytoplasmic GFP fluorescent signal for the experiment shown in (A). Error bars, SD. **(C-D)** Immunofluorescence of *ZIP1 orc1-3mAID* (C) and *zip1Δ orc1-3mAID* (D) spread meiotic chromosomes stained with anti-Pch2 (green) and anti-Nsr1 (red) antibodies, and DAPI (blue). Representative nuclei, either mock-treated, or treated with 500 ng/ml LMB 15 h after meiotic induction, are shown. Auxin (500 μM) was added to the cultures 12 h after meiotic induction to induce Orc1 degradation. Spreads were prepared at 19 h. Arrows point to the nucleolar region (nlo) and the Polycomplex (PC). Scale bar, 2 μm. The strains are the same used in (A). **(E)** Quantification of Pch2 signal for the experiment shown in (C-D). Error bars, SD; a.u., arbitrary units.

substitutions of hydrophobic residues (Leu, Ile, Val or Phe) present in each predicted putative NES to alanine (*pch2-ntd^98-107^-6A*, *pch2-ntd^127-136^-5A* and *pch2-ntd^205-214^-4A*) (S3A Fig). Centromeric plasmids containing these constructs were transformed into a *zip1Δ pch2Δ* strain and the localization of GFP-Pch2 was examined in live meiotic cells. The *pch2-ntd^98-107^-6A* and *pch2-ntd^127-136^-5A* mutations both abolished nucleolar localization and led to an exclusively cytoplasmic distribution of Pch2 (S3A and S3B Fig). These mutations lie in the extended region of the NTD required for Orc1 interaction [71], thus explaining the defective nucleolar targeting; therefore, they were discarded for further analyses. However, the GFP-Pch2-ntd^205-214^-4A version showed a strong accumulation inside the nucleus as expected if a functional

NES is disrupted (S3A and S3B Fig). Functional NESs need to be accessible to Crm1 binding and they do not usually appear within compactly folded domains. NESs tend to be present at the N terminus, at the C terminus, or within unstructured domains of exportin cargoes [72, 73]. In line with these notions, the AlphaFold structural model of Pch2 predicts that the sequences 98–107 and 127–136 are located in a highly structured region of the Pch2 NTD, whereas the residues 205–214 are displayed in a disordered and readily accessible region to support the interaction with Crm1 (S3C Fig). Therefore, we generated strains harboring the *GFP-pch2-ntd*$^{205-214}$*-4A* mutation (hereafter named as *pch2-nes4A*) integrated at the *PCH2* genomic locus for more detailed analyses. We confirmed the significant increase in nuclear localization both in the nucleolus and nucleoplasm of the Pch2-nes4A version (Fig 4A and 4B). Consistent with the fact that Pch2 nuclear buildup in *zip1Δ* cells is deleterious for the checkpoint [60], the *GFP-pch2-nes4A* mutant suppressed the meiotic block of *zip1Δ* though to a lesser extent than the *zip1Δ pch2Δ* mutant did (Fig 4C). Accordingly, analysis of Hop1 distribution on chromosome axes of *ndt80Δ*-arrested strains showed that Hop1 localization was also impaired in the *zip1Δ GFP-pch2-nes4A* mutant displaying reduced signal, but to a lesser degree compared to *zip1Δ pch2Δ* (Fig 4D and 4E). Furthermore, the levels of Hop1-T318 phosphorylation were reduced in the *zip1Δ GFP-pch2-nes4A* double mutant compared to *zip1Δ* (Fig 4F) reflecting decreased checkpoint activity. To confirm that these checkpoint defects stem from the impaired Pch2 nucleocytoplasmic transport resulting from a defective NES and not from other possible alterations in Pch2 activity and/or structure, we added an ectopic bona-fide strong NES from the PKI protein [74] generating a *GFP-NES*$^{PKI}$*-pch2-nes4A* version. Importantly, the *GFP-NES*$^{PKI}$*-pch2-nes4A* construct alleviated the nuclear accumulation of GFP-Pch2-nes4A (Fig 4A and 4B) and reestablished checkpoint function, as revealed by the restoration of the meiotic block (Fig 4C), rescue of Hop1 axial localization (Fig 4D and 4E), and higher Hop1-T318 phosphorylation levels especially at the later time point (48 h) (Fig 4F). Thus, we conclude that the $^{205}$**LSTEFDKIDL**$^{214}$ sequence in the Pch2 NTD domain (designated as NES$^{Pch2}$) behaves as a functional NES matching the Class 1a signature [70]. We demonstrate that NES$^{Pch2}$ drives the nuclear export of Pch2, and it is critical for maintaining a balance of Pch2 subcellular distribution essential for a precise meiotic recombination checkpoint response. These findings also indicate that Pch2 does not need additional partners, besides the nuclear export machinery, to travel from the nucleus to the cytoplasm where it exerts its essential checkpoint activating function.

## The *pch2-nes4A* mutation produces a minor phenotypic impact on normal *ZIP1* meiosis

We have previously shown that, like *pch2Δ*, the forced massive accumulation of Pch2 in the nucleus provoked by its fusion to a strong NLS (GFP-NLS$^{SV40}$-Pch2) has little or no impact on meiotic progression and spore viability in an otherwise unperturbed (*ZIP1*) meiosis [60]. Nonetheless, spore viability is compromised in both *pch2Δ* and *GFP-NLS*$^{SV40}$*-pch2* in a *spo11-3HA* hypomorphic mutant background that generates reduced DSB levels [37, 60, 75]. Consistent with these observations, meiotic progression and sporulation efficiency were normal in the *GFP-pch2-nes4A* single mutant (Fig 5A and 5B), and spore viability was only mildly impaired in *spo11-3HA GFP-pch2-nes4A* (Fig 5C). Analysis of spread nuclei revealed that Pch2-nes4A signal was higher, Zip1 localization was normal, and polycomplexes were not observed in *GFP-pch2-nes4A* (Fig 5D, 5F and 5G). However, we detected increased polycomplex formation in prophase-arrested *ndt80Δ GFP-pch2-nes4A* mutants compared to *ndt80Δ GFP-PCH2* (Fig 5E and 5G). These observations suggest that the increased formation of these assemblies observed upon LMB treatment (S2A–S2C Fig) may be caused by the combination

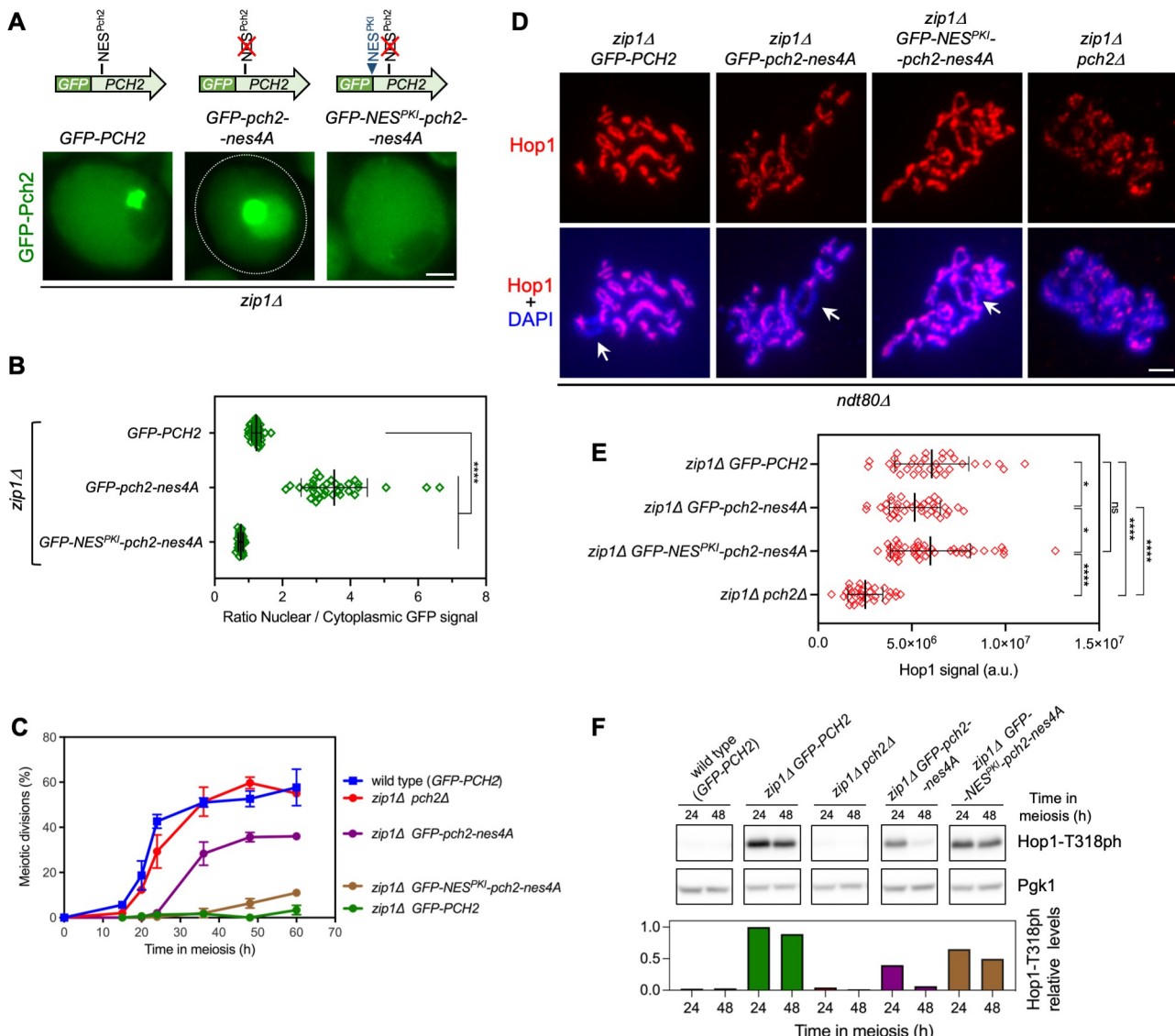

**Fig 4. A functional NES in the NTD of Pch2 promotes its nuclear export.** (A) Fluorescence microscopy images of the localization of GFP-Pch2, GFP-Pch2-nes4A and GFP-NES^PKI-Pch2-nes4A in *zip1Δ* cells. Images were taken 15 h after meiotic induction. Representative cells are shown. A schematic representation of the different GFP-Pch2 versions is also shown on top. Scale bar, 2 μm. Strains are: DP1625 (*zip1Δ GFP-PCH2*), DP1986 (*zip1Δ GFP-pch2-nes4A*) and DP1992 (*zip1Δ GFP-NES^PKI-pch2-nes4A*). (B) Quantification of the ratio of nuclear (including nucleolar) to cytoplasmic GFP fluorescent signal for the experiment shown in (A). Error bars, SD. (C) Time course analysis of meiotic nuclear divisions; the percentage of cells containing two or more nuclei is represented. Error bars: SD; n = 3. At least 300 cells were scored for each strain at every time point. Strains are: DP1624 (*GFP-PCH2*), DP1625 (*zip1Δ GFP-PCH2*), DP1029 (*zip1Δ pch2Δ*), DP1986 (*zip1Δ GFP-pch2-nes4A*) and DP1992 (*zip1Δ GFP-NES^PKI-pch2-nes4A*). (D) Immunofluorescence of spread meiotic chromosomes stained with anti-Hop1 antibodies and DAPI (blue). Representative nuclei from prophase-arrested *ndt80Δ* strains are shown. Arrows point to the rDNA region. Spreads were prepared at 24 h. Scale bar, 2 μm. Strains are: DP1655 (*ndt80Δ zip1Δ GFP-PCH2*), DP1988 (*ndt80Δ zip1Δ GFP-pch2-nes4A*), DP2003 (*ndt80Δ zip1Δ GFP-NES^PKI-pch2-nes4A*), and DP881 (*ndt80Δ zip1Δ pch2Δ*). (E) Quantification of the Hop1 signal for the experiment shown in (D). (F) Western blot analysis of Hop1-T318 phosphorylation in the strains analyzed in (C) at the indicated time points in meiosis. Pgk1 was used as a loading control. The graph shows the quantification of Hop1-T318ph relative levels.

of Pch2 nuclear accumulation with the cell cycle arrest imposed by the block of the nuclear export of other factors. We conclude that, similar to its complete absence (*pch2Δ*), altered Pch2 subcellular distribution (*pch2-nes4A*) provokes a stronger effect in the checkpoint-inducing *zip1Δ* context than in a normal *ZIP1* meiosis.

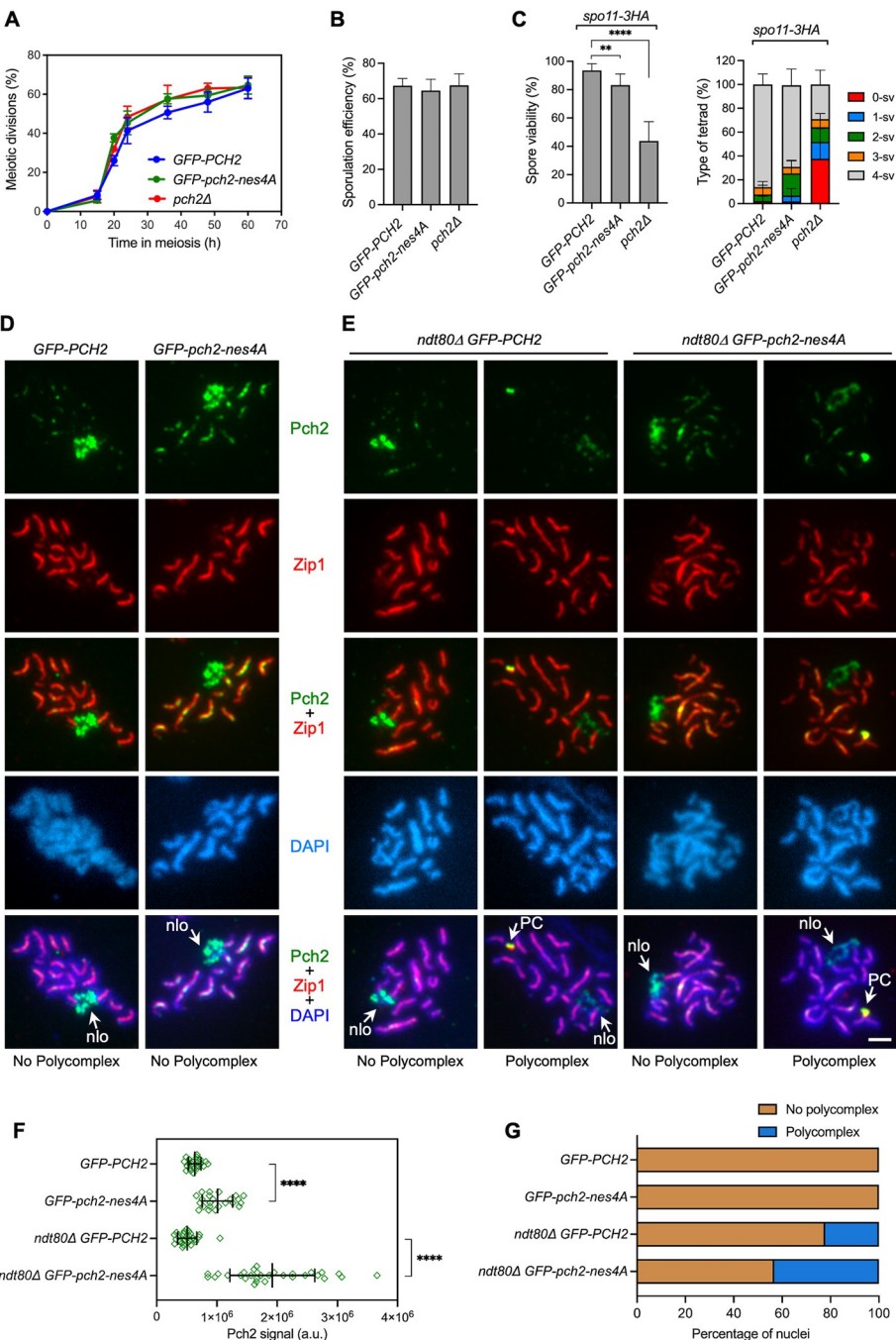

**Fig 5. The *pch2-nes4A* mutant has little impact on an unperturbed *ZIP1* meiosis. (A)** Time course analysis of meiotic nuclear divisions; the percentage of cells containing two or more nuclei is represented. Error bars: SD; n = 3. At least 300 cells were scored for each strain at every time point. **(B)** Sporulation efficiency, assessed by microscopic counting of asci, was examined after 3 days on sporulation plates. Error bars, SD; n = 3. At least 600 cells were counted for each strain. **(C)** Spore viability, assessed by tetrad dissection, is shown in the left graph. The percentage of tetrads containing 4-, 3-, 2-, 1-, and 0-viable spores is presented in the right graph. At least 144 tetrads were dissected for each strain. Error bars, SD. **(D, E)** Immunofluorescence of spread meiotic chromosomes at pachytene stained with anti-GFP antibodies (to detect GFP-Pch2; green) and anti-Zip1 antibodies (red), and DAPI (blue). Representative nuclei are shown. Spreads were prepared 15 h (D) or 24 h (E) after meiotic induction. Arrows point to the rDNA region (nlo) and polycomplex (PC). Scale bar, 2 μm. **(F)** Quantification of Pch2 signal on spread nuclei from the experiment shown in (D, E). a.u., arbitrary units. **(G)** Percentage of nuclei containing polycomplexes in the experiment shown in (D, E). Between 23 to 30 nuclei were scored in (F, G). Strains are: DP1023 (*pch2Δ*), DP1620 (*GFP-PCH2*), DP2052 (*GFP-pch2-nes4A*), DP1788 (*spo11-3HA GFP-PCH2*), DP2046 (*spo11-3HA GFP-pch2-nes4A*), DP1787 (*spo11-3HA pch2Δ*), DP1639 (*ndt80Δ GFP-PCH2*), and DP2053 (*ndt80Δ GFP-pch2-nes4A*).

## The checkpoint defects of *pch2-nes4A* are rescued by the fusion of a putative NES from human TRIP13

Although the budding yeast Pch2 is meiosis specific, the mammalian homolog TRIP13 is also expressed in somatic cells besides the germline. TRIP13 has been reported to be localized both in the nucleus and the cytoplasm in mammalian somatic cells [76–78]. The Pch2/TRIP13 protein family possesses a highly conserved C-terminal AAA+ ATPase domain and a non-catalytic NTD with a lower similarity degree. In addition, yeast Pch2 contains a non-conserved extension at the very beginning of the protein (Figs 6A and S4) that may be involved in the exclusive Pch2 nucleolar localization found in budding yeast via Orc1 interaction [62, 71]. Using the LocNES prediction tool to analyze the human TRIP13 NTD, we detected a presumptive NES (at positions 65–80) close to the region corresponding to NES$^{Pch2}$ (Fig 6A and S4). To determine whether this putative NES$^{TRIP13}$ is functionally active, we fused it to the NES$^{Pch2}$-deficient *pch2-nes4A* mutant generating a *GFP-NES$^{TRIP13}$-pch2-nes4A* construct. Addition of the proposed NES$^{TRIP13}$ partially restored cytoplasmic localization of Pch2-nes4A (Fig 6B and 6C). Likewise, the *zip1Δ GFP-NES$^{TRIP13}$-pch2-nes4A* double mutant displayed a notable meiotic block (Fig 6D) and increased levels of Hop1-T318ph at the 48-h time point (Fig 6E), consistent with a restoration of checkpoint activity. Furthermore, mutation to alanine of all NES$^{TRIP13}$ hydrophobic residues (*nes7A$^{TRIP13}$*) fail to reinstate cytoplasmic localization and checkpoint activity in *pch2-nes4A* (Fig 6B–6E). These observations are compatible with the notion that, indeed, the 65–80 amino acid fragment of TRIP13 NTD (**FL**TRN**V**QS**V**S**II**D-TE**L**) may contain a functional NES, because it is capable of substituting the requirement for NES$^{Pch2}$ to implement a balanced localization of Pch2 in the different subcellular compartments during yeast meiosis. Unlike NES$^{Pch2}$ (Figs 6F and S3C), the crystal structure of human TRIP13 [42] reveals that the putative NES$^{TRIP13}$ is located in a more structured region of TRIP13 (Fig 6G), reducing the likeness of being a real NES in the mammalian TRIP13 protein. On the other hand, it is situated in the NTD, and it is flanked by two short unstructured fragments, which are features present in some confirmed NESs located in boundary regions [73]. In any case, the fact that this 65–80 amino acid region of TRIP13 is capable of functionally substituting for NES$^{Pch2}$ when transplanted into Pch2-nes4A may be alternatively explained because the hydrophobic residues could be much more exposed in this chimeric context (Fig 6H), and it does not conclusively prove that NES$^{Pch2}$ is conserved in TRIP13. Additional studies in mammalian systems will be required to solve this question.

## Concluding remarks

Homeostatic control of Pch2 subcellular localization is crucial for a proper meiotic recombination checkpoint response [60, 61]. We unveil here another layer of control of Pch2 function during meiosis involving the nuclear export via the exportin pathway that is essential to maintain the critical balance of Pch2 distribution among different compartments (Fig 7). Functions for Pch2 outside the nucleus are not exclusive of yeast meiosis; in plants, PCH2 also operates in the cytoplasm promoting the nuclear targeting of the HORMA protein ASY1 (Hop1 homolog), in addition to its nuclear action in meiotic chromosome axis remodeling [47, 48]. It is tempting to speculate about the possibility of an evolutionary conservation of the nuclear export mechanism for Pch2/TRIP13 from yeast to mammals. In mice, TRIP13 is required for proper completion of meiotic chromosome synapsis and recombination, and *Trip13*-deficient mice are sterile [36, 79]. Although the localization of TRIP13 during mammalian meiosis has not been reported, it can be detected both in the cytoplasm and nucleus of mouse spermatocytes (I. Roig, personal communication), raising the possibility that the meiotic role of mammalian TRIP13 may also be modulated by nucleocytoplasmic transport. In humans, biallelic mutations in the *Trip13*

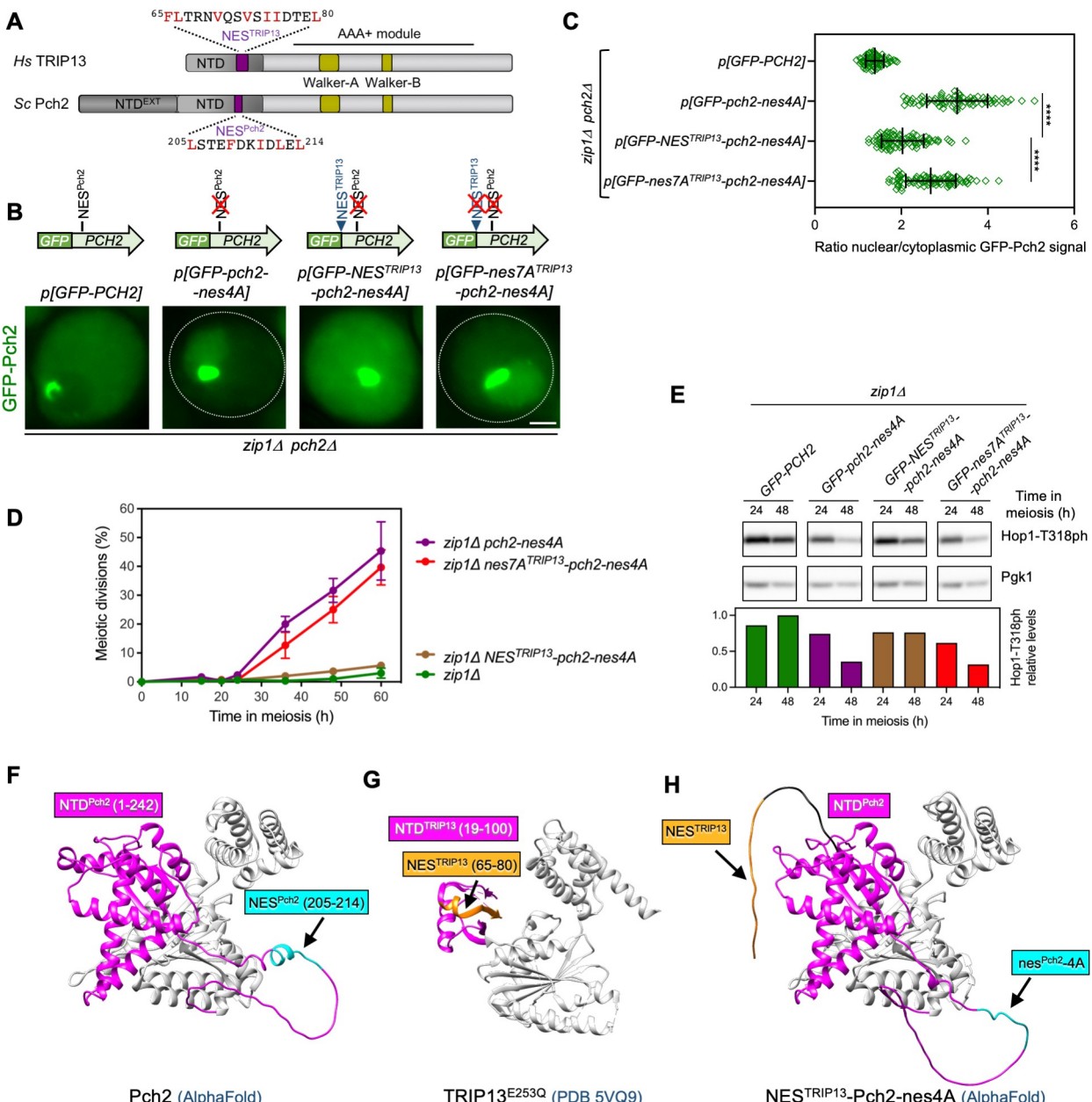

**Fig 6. Fusion of a putative NES from human TRIP13 restores checkpoint function to Pch2-nes4A. (A)** Schematic representation of the *S. cerevisiae* Pch2 protein (ScPch2) and the human ortholog (HsTRIP13) indicating the characteristic AAA+ ATPase motifs, the conserved N-terminal domain (NTD) and the extended N-terminal domain exclusive of yeast Pch2 (NTD^EXT). The positions and sequences of the putative NESs analyzed in this work are depicted (purple boxes). **(B)** Fluorescence microscopy images of the localization of GFP-Pch2, GFP-Pch2-nes4A, GFP-NES^TRIP13-Pch2-nes4A, and GFP-nes7A^TRIP13-Pch2-nes4A in *zip1Δ* cells. Images were taken 16 h after meiotic induction. Representative cells are shown. A schematic representation of the different GFP-Pch2 versions is also shown on top. Scale bar, 2 μm. The strain is DP1405 (*zip1Δ pch2Δ*) transformed with the centromeric plasmids pSS393 (*GFP-PCH2*), pSS459 (*GFP-pch2-nes4A*), pSS472 (*GFP-NES^TRIP13-pch2-nes4A*) and pSS474 (*GFP-nes7A^TRIP13-pch2-nes4A*). **(C)** Quantification of the ratio of nuclear (including nucleolar) to cytoplasmic GFP fluorescent signal for the experiment shown in (B). Error bars, SD. **(D)** Time course analysis of meiotic nuclear divisions. The percentage of cells containing two or more nuclei is represented. Error bars: SD; n = 3. At least 300 cells were scored for each strain at every time point. Strains are: DP1625 (*zip1Δ GFP-PCH2*), DP1986 (*zip1Δ GFP-pch2-nes4A*), DP2025 (*zip1Δ GFP-NES^TRIP13-pch2-nes4A*) and DP2033 (*zip1Δ GFP-nes7A^TRIP13-pch2-nes4A*). **(E)** Western blot analysis of Hop1-T318 phosphorylation in the strains analyzed in (D) at the indicated time points in meiosis. Pgk1 was used as a loading control. The graph shows the quantification of Hop1-T318ph relative levels. **(F)** Pch2 structure modeled by AlphaFold. **(G)** Human TRIP13^E253Q structure obtained from the Protein Data Bank (PDB 5VQ9) [42]. **(H)** AlphaFold predicted structure resulting from the fusion of NES^TRIP13 to Pch2-nes4A. In (F-H), the N-terminal domains of Pch2 and TRIP13 are labeled in pink, the position of NES^Pch2 (or nes^Pch2-4A) is labeled in blue, and the position of the putative NES^TRIP13 is labeled in orange.

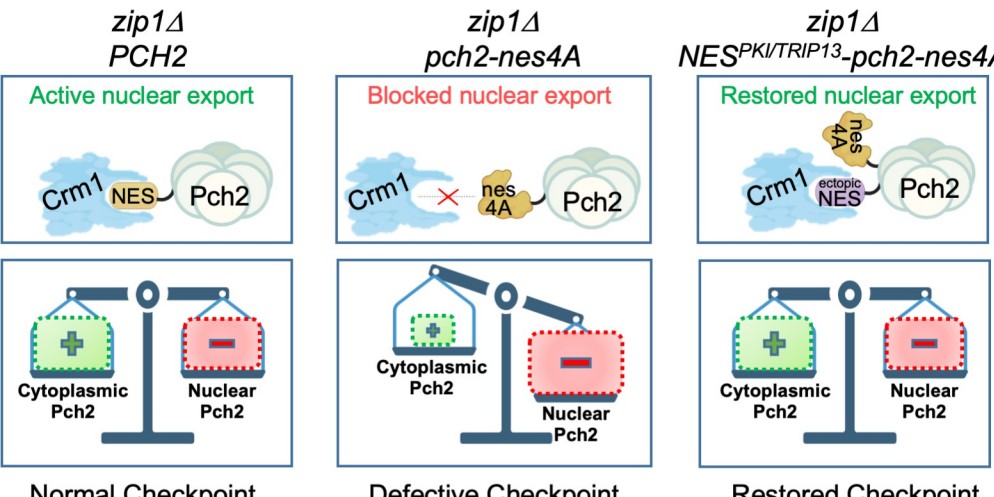

**Fig 7. NES-mediated nuclear export of Pch2 is essential for meiotic recombination checkpoint function.** The nucleocytoplasmic transport of Pch2 via the exportin pathway maintains Pch2 homeostasis and supports proper checkpoint activity in *zip1Δ*. In the NES-deficient *pch2-nes4A* mutant, the accumulation of Pch2 in the nucleus leads to checkpoint inactivation. The fusion to Pch2-nes4A of a characterized ectopic NES from the PKI protein, or a putative NES from the human TRIP13 ortholog, restores the balanced subcellular distribution of Pch2 and restores checkpoint function. For simplicity, the NES is drawn only in one subunit of the Pch2 hexamer. BioRender.com was used to create this figure.

gene are associated with female infertility and Wilms tumor [80–82]. Interestingly, the CRM1/XPO1 exportin has been recently identified in a screen for potential druggable targets involving TRIP13 function in Wilms tumor derived lines. Furthermore, the FDA-approved drug selinexor (KPT-330), which inhibits nuclear export, leads to suppression of TRIP13 function in these cell lines and it has been proposed as a potential therapeutic strategy for Wilms tumor patients [83]. In sum, all these observations underscore the potential relevance of Pch2/TRIP13 nucleocytoplasmic traffic in the multiple biological processes impacted by this protein family.

## Materials and methods

### Yeast strains

The genotypes of yeast strains are listed in S1 Table. All strains are in the BR1919 background [84]. The *zip1Δ::LEU2*, *zip1Δ::LYS2*, *ndt80Δ::kanMX6*, *pch2Δ::TRP1* and *pch2Δ::URA3* gene deletions were previously described [13, 20, 29, 85]. The *orc1-3mAID* and $P_{HOP1}$-*GFP-PCH2* constructs have been previously described [60, 62]. For the experiments involving the inactivation of the Crm1 exportin with LMB we employed diploid strains with both copies of the *CRM1* essential gene deleted (*crm1Δ::hphMX4* or *crm1Δ::natMX4*) using a PCR-based approach [86] carrying a centromeric plasmid expressing the *crm1-T539C* allele (pSS416) as the only source of exportin in the cell. The $P_{HOP1}$-*GFP-pch2-nes4A*, $P_{HOP1}$-*GFP-NES^{PKI}*-*pch2-nes4A*, $P_{HOP1}$-*GFP-NES^{TRIP13}*-*pch2-nes4A*, and $P_{HOP1}$-*GFP-nes7A^{TRIP13}*-*pch2-nes4A* constructs were introduced into the genomic locus of *PCH2* using an adaptation of the *delitto perfetto* technique [87]. Basically, PCR fragments flanked by the appropriate sequences containing the *HOP1* promoter followed by the *GFP-pch2-nes4A*, *GFP-NES^{PKI}*-*pch2-nes4A*, *GFP-NES^{TRIP13}*-*pch2-nes4A* or *GFP-nes7A^{TRIP13}*-*pch2-nes4A* sequences, and a five Gly-Ala repeat linker before the second codon of *PCH2*, were amplified from pSS459, pSS462, pSS472 and pSS474, respectively (see below). These fragments were transformed into a strain carrying the CORE cassette (*kanMX4-URA3*) inserted close to the 5' end of *PCH2*. G418-sensitive and

5-FOA-resistant clones containing the correct integrated construct, which results in the elimination of 91 nt of the *PCH2* promoter, were selected. All constructions and mutations were verified by PCR analysis and/or sequencing. The sequences of all primers used in strain construction are available in S3 File. All strains were made by direct transformation of haploid parents or by genetic crosses always in an isogenic background. Diploids were made by mating the corresponding haploid parents and isolation of zygotes by micromanipulation.

## Plasmids

The plasmids used are listed in S2 Table. The pSS393 centromeric plasmid expressing $P_{HOP1}$-*GFP-PCH2* was previously described [62]. The pSS416 plasmid containing *crm1-T539C* was kindly provided by M. Dosil (CIC, Salamanca). The pSS448 and pSS451 plasmids, driving the expression of $P_{HOP1}$-*GFP-pch2-ntd*$^{98-107}$-*6A* and $P_{HOP1}$-*GFP-pch2-ntd*$^{127-136}$-*5A*, respectively, were made by non-overlapping mutagenesis following the procedure described in the Q5 site-directed mutagenesis kit (New England Biolabs), using the pSS393 as template and divergent primers. The primers carried the sequence encoding the mutated 98–107 (**LI**RS**L**AK**VLL** to **AA**RS**A**AK**AAA**) or 127–136 (**L**F**L**S**LF**V**KKI** to **A**F**A**S**AF**A**KKA**) regions in the Pch2 NTD. The pSS459 plasmid driving the expression of $P_{HOP1}$-*GFP-pch2-nes4A* was derived from pSS393 by using the NEBuilder assembly kit (New England Biolabs) and a synthesized gBlock fragment (IDT) containing the mutated sequence corresponding to the 205–214 region (**L**STE**F**DK**I**D**L** to **A**STE**A**DK**A**D**A**). To construct the pSS462 plasmid containing $P_{HOP1}$-*GFP-NES*$^{PKI}$-*pch2-nes4A*, a 1.8-kb fragment was amplified from pSS459 ($P_{HOP1}$-*GFP-pch2-nes4A)*, using a forward primer containing the sequence encoding the Nuclear Export Signal (NES) from the PKI protein (LALKLAGLDI) [74] preceded by a *Not*I site at the 5' end, and a reverse primer within the *PCH2* coding sequence downstream of the endogenous *Blp*I site. The fragment was digested with *Not*I-*Blp*I and cloned into the same sites of pSS393. To generate the pSS472 plasmid that contains $P_{HOP1}$-*GFP-NES*$^{TRIP13}$-*pch2-nes4A*, a 418-bp fragment was amplified from pSS393 ($P_{HOP1}$-*GFP-PCH2*) using a forward primer containing the presumptive NES of TRIP13 (at positions 65–80) preceded by a *Not*I site at its 5'end, and a reverse primer within the *PCH2* coding sequence downstream of the endogenous *BamH*I site. The fragment was digested with *Not*I-*BamH*I and cloned into the same sites of pSS459. To make the pSS474 plasmid containing $P_{HOP1}$-*GFP-nes7A*$^{TRIP13}$-*pch2-nes4A*, a 418pb fragment was amplified from pSS393, using a forward primer containing the mutation to alanine of all NES$^{TRIP13}$ hydrophobic residues (**F**L**T**RN**V**QS**V**S**II**DTE**L** to **AA**TRN**A**QS**A**S**AA**DTEA), preceded by a *Not*I site at its 5'end, and a reverse primer within the *PCH2* coding sequence downstream of the endogenous *BamH*I site. The fragment was digested with *Not*I-*BamH*I and cloned into the same sites of pSS459. The sequences of primers used in plasmid construction are available in S3 File.

## Meiotic cultures and meiotic time courses

To induce meiosis and sporulation, BR strains were grown in 3.5 ml of synthetic complete medium (2% glucose, 0.7% yeast nitrogen base without amino acids, 0.05% adenine, and complete supplement mixture from Formedium at twice the particular concentration indicated by the manufacturer) for 20–24 h, then transferred to 2.5 ml of YPDA (1% yeast extract, 2% peptone, 2% glucose, and 0.02% adenine) and incubated to saturation for an additional 8 h. Cells were harvested, washed with 2% potassium acetate (KAc), resuspended into 2% KAc (10 ml), and incubated at 30˚C with vigorous shaking to induce meiosis. Both YPDA and 2% KAc were supplemented with 20 mM adenine and 10 mM uracil. The culture volumes were scaled up when needed. To inhibit Crm1-T539C, cultures were treated with 500 ng/ml of leptomycin B (LMB) at 15h in meiosis for the indicated periods of time. To induce Orc1-3mAID

degradation, auxin (500μM) was added to the cultures 12 h after meiotic induction. To score meiotic nuclear divisions, samples from meiotic cultures were taken at different time points, fixed in 70% ethanol, washed in phosphate-buffered saline (PBS) and stained with 1 μg/μl 4′,6-diamidino-2- phenylindole (DAPI) for 15 min. At least 300 cells were counted at each time point. Meiotic time courses were repeated several times; averages and error bars from at least three replicates are shown.

## Western blotting

Total cell extracts for Western blot analysis were prepared by trichloroacetic acid (TCA) precipitation from 3-ml aliquots of sporulation cultures, as previously described [88]. The antibodies used are listed in S3 Table. Antibodies were diluted in TBS 0.1% Tween containing 5% milk (for Pch2 and Pgk1), or 0.1% BSA (Bovine Serum Albumin) (for Hop1-T318ph). PVDF membranes were incubated at 4°C overnight (primary antibodies), or at room temperature for 45 minutes (secondary antibodies). The Pierce ECL Plus reagent (ThermoFisher Scientific) was used for detection. The signal was captured with a Fusion FX6 system (Vilber) and quantified with the Evolution-Capt software (Vilber).

## Cytology

Immunofluorescence of chromosome spreads was performed essentially as described [84]. Spreads were prepared at 15 h for Fig 5D, at 19 h for Figs 1F, 1G, 2A, 3C, 3D, S2A and S2B, and at 24 h for Figs 4D and 5E. The antibodies used are listed in S3 Table. Antibodies were diluted in PBS containing 3% BSA and 10% FBS (Fetal Bovine Serum). Slides were incubated at 4°C overnight (primary antibodies), or at room temperature for 2 hours (secondary antibodies) in a humid chamber. Images of spreads were captured with a Nikon Eclipse 90i fluorescence microscope controlled with MetaMorph software (Molecular Devices) and equipped with a Hammamatsu Orca-AG charge-coupled device (CCD) camera and a PlanApo VC 100x 1.4 NA objective. To measure Pch2 and Hop1 intensity on chromosome spreads, a region containing DAPI-stained chromatin was defined and the Raw Integrated Density values were measured. Background values were subtracted using the rolling ball algorithm from Fiji setting the radius to 50 pixels. Images of whole live cells expressing *GFP-PCH2* were captured with an Olympus IX71 fluorescence microscope equipped with a personal DeltaVision system, a CoolSnap HQ2 (Photometrics) camera, and 100x UPLSAPO 1.4 NA objective. Stacks of 7 planes at 0.8-μm intervals were collected. Maximum intensity projections of 3 planes containing GFP-Pch2 are shown for *ZIP1* cells and single planes for *zip1Δ* cells. To determine the nuclear/cytoplasm GFP fluorescence ratio shown in Figs 1D, 3B, 4B, 6C and S3B, the ROI manager tool of Fiji software [89] was used to define the cytoplasm and nuclear (including the nucleolus) areas; the mean intensity values were measured and subjected to background subtraction. In the specific case of Fig 6C, due to the difficulty in the discrimination of the nuclei in the strain expressing *GFP-NES^TRIP13^-pch2-nes4A*, we measured the average area of more than 50 nuclei from each one of the other strains analyzed in that Figure in which the contour of the nucleus was visible, resulting in a value of 7.02 μm². We then applied this value to draw a circle containing the nucleolus in all the cells to define the nucleus and measured the nuclear/cytoplasmic fluorescence intensity ratio.

## Sporulation efficiency and spore viability

Sporulation efficiency was quantitated by microscopic examination of asci formation after 3 days on sporulation plates. Both mature and immature asci were scored. At least 300 cells were counted for every strain. Spore viability was assessed by tetrad dissection. At least 576 spores were scored for every strain.

## Structure Analysis, software, and statistics

Yeast Pch2 (Uniprot 38126) predicted structure was downloaded from AlphaFold database (https://alphafold.ebi.ac.uk/entry/P38126) [90]. Structural conformation of human TRIP13$^{E253Q}$ Apo was downloaded from Protein Data Bank (PDB ID 5VQ9) [42]. The structure of NES$^{TRIP13}$-Pch2-nes4A was predicted using AlphaFold Colab notebook (https://AlphaFold.ipynb) based on the sequence shown in S5 Fig. The UCSF Chimera software was used to visualize and mark protein structure models [91]. To determine the statistical significance of differences, a two-tailed Student t-test was used. P-values were calculated with the GraphPad Prism 9.0 and 10.0 software. $P<0.05$ (*); $P<0.01$ (**); $P<0.001$ (***); $P<0.0001$ (****). The nature of the error bars in the graphical representations and the number of biological replicates are indicated in the corresponding figure legend.

## Supporting information

**S1 Fig. The *crm1-T539C* mutation does not substantially impair meiosis or checkpoint function in the absence of LMB. (A)** Sporulation efficiency was examined after 3 days on sporulation plates. Error bars, SD; n = 3. At least 300 cells were counted for each strain. Strains are: DP421 (wild type), DP1717 (*crm1-T539C GFP-PCH2*), DP422 (*zip1Δ*), and DP1721 (*zip1Δ crm1-T539C GFP-PCH2*). **(B)** Time course analysis of meiotic nuclear divisions. The percentage of cells containing two or more nuclei is represented. Ethanol (Mock) or Leptomycin B (LMB) were added 15 h after meiotic induction (arrow). Error bars: SD; n = 3. At least 300 cells were scored for each strain at every time point. Strains are: DP1620 (*ZIP1 CRM1 GFP-PCH2*), DP1717 (*ZIP1 crm1-T539C GFP-PCH2*), DP1621 (*zip1Δ CRM1 GFP-PCH2*) and DP1721 (*zip1Δ crm1-T539C GFP-PCH2*).
(TIF)

**S2 Fig. Nuclear accumulation of Pch2 is linked to increased association of Pch2 with the SC and assemblies of SC components in *ZIP1+* cells. (A-B)** Immunofluorescence of spread meiotic chromosomes at pachytene stained with anti-GFP antibodies (to detect GFP-Pch2; green), anti-Zip1 antibodies (red) and DAPI (blue). Representative nuclei are shown. In both, (A) and (B), cultures were mock-treated, or treated with 500 ng/ml LMB 15 h after meiotic induction. In (B), Auxin (500μM) was also added 12 h after meiotic induction to degrade Orc1. Spreads were prepared at 19 h. Arrows point to the rDNA region (nlo) and Polycomplex (PC). Scale bar, 2 μm. The strain in (A) is: DP1717 (*crm1-T539C GFP-PCH2*). The strain in (B) is: DP1885 (*orc1-3mAID crm1-T539C GFP-PCH2*). **(C)** Percentage of nuclei containing polycomplexes in the experiments shown in (A and B). Between 10 to 20 nuclei were counted for each strain and condition. **(D)** Quantification of the different patterns of Pch2 localization in the experiment presented in Fig 3A and 3B. Representative cells are shown.
(TIF)

**S3 Fig. Identification of NES sequences in Pch2. (A)** Schematic representation of the *S. cerevisiae* Pch2 protein. The position and sequence of the three putative NESs predicted by LocNES in the non-catalytic N-terminal domain of Pch2 are depicted, as well as the corresponding mutants generated. The images show representative *zip1Δ pch2Δ* cells transformed with centromeric plasmids expressing wild-type *GFP-PCH2* or the different mutated versions of the predicted NESs, as indicated. Note that only the mutation of the 205–214 region (boxed) leads to Pch2 accumulation in the nucleus. Images were taken 15 h after meiotic induction. The strain is DP1405 (*zip1Δ pch2Δ*) transformed with the centromeric plasmids pSS393 (*GFP-PCH2*), pSS448 (*GFP-pch2-ntd$^{98-107}$-6A*), pSS451 (*GFP-pch2-ntd$^{127-136}$-5A*) and pSS459 (*GFP- pch2-ntd$^{205-214}$-4A*). **(B)** Quantification of the ratio of nuclear (including nucleolar) to

cytoplasmic GFP fluorescent signal for the experiment shown in (A). Error bars, SD. **(C)** AlphaFold model of Pch2 structure. The N-terminal domain of Pch2 is labeled in pink. The positions of the putative NESs analyzed are labeled in yellow (98–107 region), green (127–136) and blue (205–214).
(TIF)

**S4 Fig. Sequence position of the NES identified in Pch2 and the putative NES in TRIP13.** ClustalW alignment of the protein sequences of Pch2 orthologs from *S. cerevisiae* (ScPch2) and human (HsTRIP13). The characteristic AAA+ ATPase features are boxed in blue. The presumed NESs analyzed are boxed in red. The color code for amino acids is the following: AVFP-MILW (small + hydrophobic -Y): red. DE (acidic): blue. RK (basic -H): magenta. STYHCNGQ (hydroxyl +sulfhydryl + amine + G): green. Alignment was performed at https://www.ebi.ac.uk/Tools/msa/clustalo/.
(TIF)

**S5 Fig. Protein sequence of NES^TRIP13^-Pch2-nes4A.** Sequence used for the AlphaFold prediction of NES^TRIP13^-Pch2-nes4A structure shown in Fig 6H. Relevant motifs are indicated. Pch2 N-terminal domain (NTD) is in pink color.
(PDF)

**S1 Table. *Saccharomyces cerevisiae* strains.**
(PDF)

**S2 Table. Plasmids.**
(PDF)

**S3 Table. Antibodies.**
(PDF)

**S1 File. Raw data.** Excel workbook with separate spreadsheets containing numerical data underlying the corresponding figure panels.
(XLSX)

**S2 File. Statistics summary.**
(XLSX)

**S3 File. Primers.** Sequences of oligonucleotides used in the construction of strains and plasmids.
(XLSX)

## Acknowledgments

We are grateful to Andrés Clemente for helpful comments and discussions and to Ignasi Roig for sharing unpublished results. We also thank Mercedes Dosil and Yolanda Sánchez for reagents, Jesús Pinto, Carmen Castro and Carlos R. Vázquez for advice on microscopy analysis, Isabel Acosta for technical support, Ana Lago-Maciel for help in strain construction, and Lydia Iglesias for advice on AlphaFold predictions.

## Author Contributions

**Conceptualization:** Esther Herruzo, Estefanía Sánchez-Díaz, Beatriz Santos, Jesús A. Carballo, Pedro A. San-Segundo.

**Formal analysis:** Esther Herruzo, Estefanía Sánchez-Díaz, Sara González-Arranz.

**Funding acquisition:** Jesús A. Carballo, Pedro A. San-Segundo.

**Investigation:** Esther Herruzo, Estefanía Sánchez-Díaz, Sara González-Arranz, Pedro A. San-Segundo.

**Project administration:** Beatriz Santos.

**Resources:** Jesús A. Carballo.

**Supervision:** Beatriz Santos, Pedro A. San-Segundo.

**Visualization:** Esther Herruzo, Estefanía Sánchez-Díaz, Pedro A. San-Segundo.

**Writing – original draft:** Pedro A. San-Segundo.

**Writing – review & editing:** Pedro A. San-Segundo.

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
