## [Decision Letter · Decision Letter 0]

19 Aug 2023

Dear Pedro,

Thank you very much for submitting your Research Article entitled 'Exportin-mediated nucleocytoplasmic transport maintains Pch2 homeostasis during meiosis' to PLOS Genetics.

The manuscript was fully evaluated at the editorial level and by independent peer reviewers. The reviewers appreciated the attention to an important topic but identified some concerns that we ask you address in a revised manuscript.

We therefore ask you to modify the manuscript according to the review and academic editor recommendations. Your revisions should address the specific points made by each reviewer.

1) Provide a detailed list of your responses to the review comments and to the academic editor's comments and a description of the changes you have made in the manuscript.

Yours sincerely,

Michael Lichten, Ph.D.

Academic Editor

PLOS Genetics

Gregory P. Copenhaver

Editor-in-Chief

PLOS Genetics

Academic Editor's comments:

1. Line 162. While normal sporulation efficiency is shown for crm1-T539C, meiotic kinetics is not. Please include a CRM1 control so that the timing of nuclear divisions can be compared.

2. Line 176-179. Since the current work does not include a nucleolar marker for this experiment, this sentence needs to be reworded to indicate that this identification was made previously. Suggest “in mock-treated wild-type ZIP1 cells, Pch2 localized to one side of the nucleus in a region that was previously shown to correspond to the nucleolus (reference),”

and in following text (i.e. description of zip1∆ experiments) please also make it clear that this area is presumed to be the nucleolus because it was shown to be so in previous experiments, provide references, etc.

3. Lines 222 and following. These experiments do not measure the meiotic recombination checkpoint—they examine the meiotic response to double strand breaks (i.e. DSB-provoked phosphorylation of Hop1 by DNA damage response kinases). It is not possible to measure actual checkpoint activity because of the confounding effects of LMB. Therefore, please reword to indicate that what is being detected is the DNA damage response, not the checkpoint itself. Perhaps you could refer to the later results in Figure 4H in support of the conclusion that Pch2 accumulation in the nucleus interferes with the checkpoint itself. Please note, however, that this is a complicated conclusion; reduced Hop1 levels on chromosomes in cells with increased nuclear Pch2 might also reduce DSB formation, which would also compromise the checkpoint. Unless it can be shown that DSB levels are unaffected, the conclusions should be softened.

4. Making underlying data available only after the paper is accepted is not an option, because the underlying data need to be reviewed to make sure that they are provided in an appropriate format, etc. For example, underlying data for graphs that show ratios (i.e. nuclear/cytoplasmic) should be provided as original data, not as already-calculated ratios. So, please include the data underlying graphs in the next revision.

Reviewer's Responses to Questions

**Comments to the Authors:**

Reviewer #1: The manuscript by Herruzo provides compelling evidence that the cytoplasmic localization of Pch2 which the authors have previously published depends upon the active export of Pch2 from the nucleus using the Crm1 protein. Furthermore they have identified a nuclear export sequence in the amino terminus of Pch2, as well as its mammalian ortholog Trip13. By manipulating Pch2 localization through inhibition of Crm1 binding by LMB or deleting/adding nuclear export signal to the Pch2 N-terminus, they built on their previous work showing that when Pch2 accumulates in the nucleus, Hop1 is decreased on chromosomes and checkpoint activity is comprised. The experiments are well controlled and very rigorously done and, especially with the connection to Trip13, will be of interest to the meiosis community. My only comments are minor ones meant to correct some English language mistakes or improve the clarity of the presentation.

Minor comments

Line 39: “focalized” is not a word. Localized could be used instead

Line 49: traffic should be trafficking

Line 50: …involving nuclear export via…

Line 58-59…it would helpful if the words were written in the temporal order they occur in a yeast cell: pairing, recombination and synapsis.

Line 65: define the LINC acronym

Line 70, the authors are correct that in the context of the SC the protein cores containing Hop, Red1 and Rec8 are referred to as lateral elements. But in the abstract and throughout the paper this term is never used again. Instead, Hop1 is referred to as an axial protein—which is also correct. To help a broader audience, explicitly say that lateral and axial refer to the same structures, just in the presence or absence of the central region.

Line 105: two papers that should be cited for the fact that Hop1 interacts with Red1 are de los Santos and Hollingsworth, JBC 1999 and Bailis and Roeder Cell 2000.

Line 204: only associated with the rRNA region…

Line 222: It would be helpful to a general audience to explain that in zip1∆ mutants the absence of the central region does not prevent crossovers from connecting the homologous axial elements. This explains why the unsynapsed homologs are connected in Figure 2A.

Line 226, In Figure 2A +LMB, the intensity of Hop1 staining is clearly less than without the drug. But the picture shown does not support the statement that the axial staining is less continuous.

Line 226, “To elude” is an incorrect use of this word. “To prevent” or “To circumvent” would be better.

Line 240: write out nuclear localization signal instead of using the acronym

Line 241: …arguing that simply blocking Pch2 nuclear export…

Line 284: don’t use undefined acronyms in the title. Also, “drives” is too active a verb—it suggests the NES provides the motive force for nuclear export. Say instead, “A nuclear export sequence in the amino-terminus of Pch2 promotes export out of the nucleus”.

Line 286, for more concise writing, avoid unnecessary words such as “It is well known…”. Say instead, Crm1 exportin binds proteins possessing nuclear export signals…:

Line 324: delete “exquisite” as this is editorializing and subjective in the results

Line 343: delete “remarkably”, again this is editorializing in the results.

Figure legend 1: for clarity, define the LMB acronym in panel A. What are the cytoplasmic tails hanging off the nuclear pore?

For Figure 5A, the label for the middle panel could be PCH2 + LMB or pch2-NES. This would incorporate the findings presented by the authors in the first part of the paper in their model.

For Table S2, Change “Relevant parts” to Yeast genotype

Table S3, The authors should add the sources of the secondary antibodies they used and the dilutions, as well as the incubation conditions for both the primary and secondary antibodies.

In the Bibiography, only the first word of the title and proper nouns should be capitalized.

Reviewer #2: The is a straightforward characterization of the regulation of Pch2’s nuclear export in budding yeast and provides important information about the balance required for the nuclear import and export of PCH2 to regulate the recombination checkpoint and meiotic progression. The authors demonstrate that Pch2’s nuclear export requires the conserved export factor Crm1, identify the nuclear export signal in Pch2, demonstrate that nuclear export is important to maintain the meiotic recombination checkpoint and show that the signal for export on Pch2/TRIP13 is conserved between budding yeast and mammals. This is a relevant manuscript for the meiosis field, provides important insight about the regulation of Pch2 and is a rigorous study. In particular, I appreciated the authors rigor in adding the PKI NES to verify that the mutation of Pch2’s NES did not effect the protein’s function. I have two major concerns about what I think are important controls and some minor concerns that should be addressed before publication to make the paper more accessible.

Major concerns:

What are the consequences of mutating Pch2’s NES on meiosis in a strain with functional ZIP1? In particularly, what does spore viability, meiotic progression, and polycomplex formation look like? This might be be useful to determine whether the specific phenotypes observed when blocking all nuclear export (Figures S1 and S2) are because of defects in exporting other factors or defects because of exporting PCH2 specifically. Also, given that Pch2 ensures Hop1 availability for loading onto meiotic chromosomes, it may address whether enforced nuclear accumulation of Pch2 has consequences for normal meiotic progression.

What does Hop1 loading look like in zip1Δ GFP-pch2-nes4A ndt80 mutants? Does this correlate with the inability of this mutant to fully restore meiotic progression in zip1Δ? If not, this may also support the possibility that enforced nuclear accumulation of Pch2 has consequences for normal meiotic progression.

Minor concerns:

The authors use a mutation in crm1 and drug treatment to abrogate nuclear export. Is there a reason they do not use a null mutation in crm1? This information may be useful with those unfamiliar with the details of this field.

Can the authors outline the nucleus in figures where they are comparing nuclear and cytoplasmic localization of GFP-Pch2? This would be relevant for Figures 1B and C, 3A, and 4A.

The authors mention several examples where they use ndt80 null mutants to avoid cell cycle timing as a possible confounder of their analysis. In examples where they are not using ndt80 mutants, can they mention in text the timepoints at which the spreads in 1F and G, 4A, C and D and their quantifications (Figures 1H-J, 4B and 4D) were performed? This information is present in the figure legends but I think it would also be useful in the text.

Is there chromosomal associated GFP-Pch2 in the mock treated zip1^ spreads (Figure 1G)? There appears to be Pch2 signals not at the nucleolus that are not present in the LMB rated zip1^ spreads.

Line 201-202: “the accumulation of Pch2 in the nucleus was also associated to the increased formation of polycomplexes” should be “the accumulation of Pch2 in the nucleus was also associated with the increased formation of polycomplexes”

Line 204: “Pch2 is only associated to the rDNA region” should be “Pch2 is only associated with the rDNA region”

Line 241: “arguing that the simply block of Pch2 nuclear export by itself may account for the impaired checkpoint activity observed in LMB-treated zip1Δ cells.” should be “arguing that simply blocking Pch2 nuclear export by itself may account for the impaired checkpoint activity observed in LMB-treated zip1Δ cells.”

Line 304: italicize GFP-pch2-ntd205-214-4A or capitalize GFP-Pch2-ntd205-214-4A to indicate the protein

Reviewer #3: Successful meiosis depends on the proper control of programmed DNA double strand break formation, which depends largely on the proper localization of the meiotic regulator Hop1. Hop1’s localization to, and eventual removal from, meiotic chromosomes relies on the AAA+ ATPase Pch2 (TRIP13 in mammals), which remodels Hop1 and promotes its dynamic relocalization at different stages of meiotic prophase. In earlier work, the San-Segundo group has demonstrated that Pch2’s localization in both the cytoplasm and the nucleus is important for its functions. In the current manuscript “Exportin-mediated nucleocytoplasmic transport maintains Pch2 homeostasis during meiosis,” Herruzo et al. identify a nuclear export signal (NES) in Pch2 and show that nuclear export is important for its roles in Hop1 regulation (and meiotic checkpoint function). The work is well done, the conclusions are clear, and I generally support publication after the authors address a few outstanding issues (detailed below).

Major concerns:

I’m not sure I agree with the conclusions in the section “Pch2 nucleocytoplasmic traffic is independent of Zip1 and Orc1.” The data in this section are clear, and the experiments are important. But the presence of Zip1 protein clearly does have an effect on the overall nuclear/cytoplasmic ratio of Pch2 - for example, compare the two purple datasets in Figure 3B, or in Figure 1D.

Related to the above, I think it would be important to compare +/- auxin data for the experiments in Figure 3A-3B. This would likely show that the presence or absence of Orc1 protein also has an effect on the nuclear/cytoplasmic ratio of Pch2.

Thus, the difference between the authors’ interpretation of these data and my own is one of emphasis: while the authors are correct that Pch2’s transport per se does not involve Zip1 or Orc1, these proteins clearly do tend to retain Pch2 in the nucleus and therefore affect the intracellular distribution of Pch2.

Regarding the section describing NES identification, the authors should consider including supporting information from the AlphaFold structural model of Pch2. This model (available on the AlphaFold database web site) shows that residues 98-107 and 127-136 are in strongly-predicted alpha helices within the Pch2 N-terminal domain. Whereas, residues 205-214 are within a region that is predicted to be disordered in solution - perfect for an NES. This is nicely supportive and the authors may wish to mention it.

I am very skeptical that the authors are testing what they think they are testing in the final Results section, where they transplant a putative NES from mammalian TRIP13 into Pch2 and observe rescue of some phenotypes. The structure of TRIP13 shows that this region (residues 65-80) is part of the structured TRIP13 N-terminal domain, and the hydrophobic residues in this region are largely buried in this domain. Thus, this sequence is highly unlikely to mediate nuclear export in the mammalian protein. When grafted onto Pch2, this stretch of sequence may well serve as a nuclear export sequence, simply because of the presence of solvent-exposed hydrophobic residues. This does not, however, prove the authors’ assertion that the NES of Pch2/TRIP13 is evolutionarily conserved. I suggest either removing this section entirely or re-framing it significantly. Much of the “Concluding Remarks” would also have to be altered if this section were removed or re-framed.

Minor points:

Figure 2A - I think it would be helpful to have panels with each channel for this figure

Figure 4B & 4G - color datasets in this panel consistent with the genotype coloring in other panels?

**Have all data underlying the figures and results presented in the manuscript been provided?**

Reviewer #1: **No: **The authors say they will provide the data underlying their figures when the paper is accepted.

Reviewer #2: None

Reviewer #3: Yes

PLOS authors have the option to publish the peer review history of their article (what does this mean?). If published, this will include your full peer review and any attached files.

Reviewer #1: No

Reviewer #2: No

Reviewer #3: No

---

## [Decision Letter · Decision Letter 1]

23 Oct 2023

Dear Pedro,

We are pleased to inform you that your manuscript entitled "Exportin-mediated nucleocytoplasmic transport maintains Pch2 homeostasis during meiosis" has been editorially accepted for publication in PLOS Genetics. Congratulations!

Before your submission can be formally accepted and sent to production you will need to complete our formatting changes, which you will receive in a follow up email. At this time, you may wish to add the reference requested by one of the reviewers, and correct the typo noticed by another reviewer. Please be aware that it may take several days for you to receive this email; during this time no action is required by you. Please note: the accept date on your published article will reflect the date of this provisional acceptance, but your manuscript will not be scheduled for publication until the required changes have been made.

Yours sincerely,

Michael Lichten, Ph.D.

Academic Editor

PLOS Genetics

Gregory P. Copenhaver

Editor-in-Chief

PLOS Genetics

Comments from the reviewers (if applicable):

Reviewer's Responses to Questions

**Comments to the Authors:**

Reviewer #1: The authors have adequately addressed all of my comments from the original draft. My only new comment is that a reference should be provided for the statement they added on line 92 that says that chromosomes in zip1∆ diploids are still connected by axial associations.

Reviewer #2: The authors have appropriately addressed my concerns in their revision.

Reviewer #3: The authors have done a good job addressing my comments and questions. Although I still disagree that the putative NES the authors identify in the TRIP13 NTD is likely to function as an NES in TRIP13, the authors do a good job of discussing the nuances of this point. I found one typo in the newly-written portion: Page 13, line 399: “Does not conclusively proof” should be “Does not conclusively prove”. Other than that, I support acceptance.

**Have all data underlying the figures and results presented in the manuscript been provided?**

Reviewer #1: Yes

Reviewer #2: Yes

Reviewer #3: Yes

PLOS authors have the option to publish the peer review history of their article (what does this mean?). If published, this will include your full peer review and any attached files.

Reviewer #1: No

Reviewer #2: No

Reviewer #3: No

**Data Deposition**

http://datadryad.org/submit?journalID=pgenetics&manu=PGENETICS-D-23-00808R1

**Press Queries**

---

## [Editor Report · Acceptance letter]

6 Nov 2023

PGENETICS-D-23-00808R1 

Exportin-mediated nucleocytoplasmic transport maintains Pch2 homeostasis during meiosis 

Dear Dr San-Segundo, 

We are pleased to inform you that your manuscript entitled "Exportin-mediated nucleocytoplasmic transport maintains Pch2 homeostasis during meiosis" has been formally accepted for publication in PLOS Genetics! Your manuscript is now with our production department and you will be notified of the publication date in due course.

With kind regards,

Anita Estes

PLOS Genetics

On behalf of:
